# Eldecalcitol Induces Minimodeling-Based Bone Formation and Inhibits Sclerostin Synthesis Preferentially in the Epiphyses Rather than the Metaphyses of the Long Bones in Rats

**DOI:** 10.3390/ijms25084257

**Published:** 2024-04-11

**Authors:** Tomoka Hasegawa, Tomomaya Yamamoto, Hiromi Hongo, Tsuneyuki Yamamoto, Mai Haraguchi-Kitakamae, Hotaka Ishizu, Tomohiro Shimizu, Hitoshi Saito, Sadaoki Sakai, Kenji Yogo, Yoshihiro Matsumoto, Norio Amizuka

**Affiliations:** 1Ultrastructure of Hard Tissue, Graduate School of Dental Medicine, Faculty of Dental Medicine, Hokkaido University, Sapporo 060-8586, Japan; tomomaya@den.hokudai.ac.jp (T.Y.); hiromi@den.hokudai.ac.jp (H.H.); haraguchi@den.hokudai.ac.jp (M.H.-K.); iszhtk24@gmail.com (H.I.); amizuka@den.hokudai.ac.jp (N.A.); 2Department of Dentistry, Japan Ground Self-Defense Force, Camp Shinmachi, Takasaki 370-1394, Japan; 3Oral Functional Anatomy, Graduate School of Dental Medicine, Faculty of Dental Medicine, Hokkaido University, Sapporo 060-8586, Japan; yamatsu@den.hokudai.ac.jp; 4Orthopedics, Graduate School of Medicine, Hokkaido University, Sapporo 060-8638, Japan; simitom@wg8.so-net.ne.jp; 5Chugai Pharmaceutical Co., Ltd., Tokyo 103-8324, Japan; saitohts@chugai-pharm.co.jp (H.S.); sakaisdo@chugai-pharm.co.jp (S.S.); yogoknj@chugai-pharm.co.jp (K.Y.); matsumotoyosh@chugai-pharm.co.jp (Y.M.)

**Keywords:** eldecalcitol, sclerostin, minimodeling, bone remodeling, osteoporosis

## Abstract

This study aimed to examine minimodeling-based bone formation between the epiphyses and metaphyses of the long bones of eldecalcitol (ELD)-administered ovariectomized rats. Sixteen-week-old female rats were divided into four groups: sham-operated rats receiving vehicle (Sham group), ovariectomized (OVX) rats receiving vehicle (Vehicle group), or ELDs (30 or 90 ng/kg BW, respectively; ELD30 and ELD90 groups). ELD administration increased bone volume and trabecular thickness, reducing the number of osteoclasts in both the epiphyses and metaphyses of OVX rats. The Sham and Vehicle groups exhibited mainly remodeling-based bone formation in both regions. The epiphyses of the ELD groups showed a significantly higher frequency of minimodeling-based bone formation than remodeling-based bone formation. In contrast, the metaphyses exhibited significantly more minimodeling-based bone formation in the ELD90 group compared with the ELD30 group. However, there was no significant difference between minimodeling-based bone formation and remodeling-based bone formation in the ELD90 group. While the minimodeling-induced new bone contained few sclerostin-immunoreactive osteocytes, the underlying pre-existing bone harbored many. The percentage of sclerostin-positive osteocytes was significantly reduced in the minimodeling-induced bone in the epiphyses but not in the metaphyses of the ELD groups. Thus, it seems likely that ELD could induce minimodeling-based bone formation in the epiphyses rather than in the metaphyses, and that ELD-driven minimodeling may be associated with the inhibition of sclerostin synthesis.

## 1. Introduction

During growth, bone development is accompanied by changes in bone shape and size by “modeling”, a process where bone formation and bone resorption take place independently [1,2]. Once an individual reaches adulthood, the bone is renewed without changes in shape or size through “bone remodeling” [1,3,4,5]. Bone remodeling, unlike modeling, is achieved by the histologically sequential processes of bone resorption and bone formation at the same spatial location. Briefly, osteoblasts deposit new bones at the exact sites where osteoclasts have previously been resorbed [1,3,4], resulting in the replacement of old bone with new bone through a balanced and coupled organization of bone resorption and bone formation by osteoclasts and osteoblasts [6]. It has been reported that modeling-based bone formation takes place during development and growth, while bone remodeling-based bone formation is predominant in the adult stage [7]. Bone remodeling in healthy adults is a crucial process for adjusting the bone architecture to maintain bone strength. In osteoporosis, however, decreases in bone mass and deteriorations in bone structure are observed.

Eldecalcitol (ELD), an analog of 1α, 25-dihydroxyvitamin D3 (1α, 25(OH)_2_D_3_), functions as an anti-resorptive reagent in the treatment of osteoporosis by suppressing osteoclastic bone resorption and enhancing bone strength in rodents [8,9] and in humans [10,11,12]. ELD possesses a 3-hydroxypropoxy moiety at the position of 2β of 1α, 25(OH)_2_D_3_, and therefore, it has been speculated that ELD may have a specific effect on the bone [13]. We have discovered that ELD increases bone volume in ovariectomized (OVX) rats by means of not only reducing osteoclastic bone resorption but also increasing the formation of focally convex bone bulges [14,15], which constitutes the process of “minimodeling” originally proposed by Frost [4].

Minimodeling is a phenomenon featuring a new focal bone on trabeculae that is formed independently of osteoclastic bone resorption; quiescent osteoblasts, also referred to as bone lining cells, are activated into mature osteoblasts, which then deposit new bone onto the existing bone. Unlike remodeling, minimodeling changes the shape and size of the trabeculae by depositing new bone onto the pre-existing bone, consequently generating a focally convex bulge and smooth arrest lines at the boundaries with the old bone [4,16,17]. Thus, minimodeling should be classified as a microscopic modeling that takes place on trabeculae but not on cortical bone [4,7,17]. It is likely that bone remodeling and minimodeling occur on the trabeculae. However, the frequency of bone turnover and mechanical loading in the trabeculae differ between the regions of the long bone, i.e., epiphysis or metaphysis [18,19]. Because of the high bone turnover and low mechanical loading in the metaphyseal trabeculae compared to the epiphyseal trabeculae, the region of metaphysis undergoes marked bone loss after ovariectomy [20,21]. As well as bone remodeling, we speculated that minimodeling induced on the trabeculae may also differ depending on the site of the long bones. However, which regions of ovariectomized long bones would preferentially induce minimodeling-based bone formation upon ELD treatment is unknown.

Recently, the anti-sclerostin antibody has been highlighted as an anabolic reagent for the treatment of osteoporosis [22,23,24,25]. The anti-sclerostin antibody reportedly stimulates osteoblastic activity and increase bone volume through modeling-based bone formation [22,23]; the anti-sclerostin antibody could increase bone volume in both trabecula and cortical bone [23]. In contrast, ELD has been reported to induce modeling in the trabeculae, i.e., minimodeling, but not on the cortical bone [14]. Therefore, it seems possible that ELD-driven minimodeling of the trabecular bone is, at least in part, related to sclerostin synthesis.

This study aimed to examine whether epiphyses or metaphyses preferentially induce minimodeling-based bone formation, and also to explore whether ELD administration would affect sclerostin synthesis in minimodeling-based bone formation, using the in vivo model of ELD-administered OVX rats.

## 2. Results

### 2.1. Femoral BMD and Bone Histomorphometrical Analyses of the Tibial Metapsyses and Epiphyses in Sham, Vehicle, ELD30, and ELD90 Groups

The BMDs of both the distal region and the whole femora in the Sham, ELD30, and ELD90 groups were significantly higher than those in the Vehicle group (Figure 1A,B).

Histologically, the trabecular number and thickness of the metaphyses in the Vehicle group seemed to be markedly decreased, whereas those in the ELD30 and ELD90 groups increased (Figure 1C–J). ELD30 and ELD90 groups exhibited thickened trabeculae in both the metaphyses and the epiphyses compared with the Sham and Vehicle groups (Figure 1G–N). The metaphyses in the Vehicle group exhibited decreases in BV/TV and Tb.N but not in Tb.Th compared with those in the Sham group (Figure 1O–Q). However, the BV/TV, Tb.N, and Tb.Th of the metaphyses in ELD90 groups were higher than those in the Vehicle group. Unlike the metaphyses, the BV/TV, Tb.N, and Tb.Th in the epiphysis did not differ between the Sham and Vehicle groups (Figure 1R–T). The epiphyseal BV/TV in the ELD30 and ELD90 groups and the Tb.Th in the ELD90 group were significantly higher than those in the Sham and Vehicle groups (Figure 1R,T). Thus, the OVX operation resulted in significant reductions in BV/TV and Tb.N in the metaphyses but not in the epiphyses, whereas ELD administration increased BV/TV and Tb.Th in both the metaphyses and the epiphyses.

### 2.2. The Distribution of TRAPase-Reactive, Cathepsin K-Positive, and ED1-Immunoreactive Osteoclasts, and ALPase-Positive Osteoblasts in the ROI of Metaphysis and Epiphysis

Histochemical assessment demonstrated that the numbers of TRAPase-reactive, cathepsin K-positive, and ED1-immunoreactive osteoclasts/macrophages in the metaphyses were elevated in the Vehicle group but decreased in the ELD groups (Figure 2A–L). Consistently, the numbers of TRAPase-reactive, cathepsin K-positive, and ED1-immunoreactive cells in the ELD90 group were significantly lower than those in the Vehicle group (Figure 2Q–S). Although the immunohistochemical analysis revealed a trend of reduced ALPase-reactive osteoblastic area in the Vehicle group (Figure 2M–P), no statistically significant difference was detected in the ALPase-positive osteoblastic areas among the groups (Figure 2T).

In the epiphysis, histochemical images demonstrated many TRAPase-reactive, cathepsin K-positive, and ED1-positive cells in the Vehicle group (Figure 3A–L). The numbers of osteoclasts positive for TRAPase, cathepsin K, and ED1 in the ELD groups were significantly lower than those in the Vehicle group (Figure 3Q–S). However, unlike the metaphysis, the ALPase-reactive osteoblastic areas seemed to be histochemically evident in the ELD30 and ELD90 groups (Figure 3M–P). The area of the ALPase-reactive regions in the ELD90 group was larger than that in the Sham group (Figure 3T).

### 2.3. Minimodeling-Based Bone Formation and Remodeling-Based Bone Formation in Tibial Epiphyses and Metaphyses

By calcein/tetracycline labeling, focally convex structures, indicative of minimodeling, were observed on the trabeculae of both the metaphyses and epiphyses in the ELD30 and ELD90 groups, but were rarely seen in the Sham and Vehicle groups (Figure 4A–P). There were seemingly more sites of minimodeling in the epiphyses compared with the metaphyses in the ELD90 group (Figure 4D,H,L,P). In both the metaphyses and epiphyses of the Sham and Vehicle groups, the percentage of remodeling-based bone formation was significantly higher than that of minimodeling-based bone formation (Figure 4Q,R). In the ELD30 group, the percentage of minimodeling-based bone formation, rather than remodeling-based bone formation, was elevated in the epiphyses but not in the metaphyses (Figure 4Q,R). In the ELD90 group, the percentage of minimodeling-based bone formation was significantly higher than remodeling-based bone formation in the epiphyses. Although the metaphyses showed significantly higher minimodeling-based bone formation in the ELD90 group compared with ELD30 group, there were no statistically significant differences between minimodeling-based bone formation and remodeling-based bone formation even in the ELD90 group. Thus, it seems likely that ELD-driven minimodeling would be preferentially induced in the epiphyses rather than in the metaphyses.

### 2.4. The Distribution of Sclerostin-Positive Osteocytes in Minimodeling-Induced New Bone

Immunohistochemical detection of ALPase and sclerostin revealed only a small number of sclerostin-immunoreactive osteocytes in the newly formed bone in both the metaphyses and the epiphyses (Figure 5A–P). However, the underlying pre-existing bone showed many sclerostin-positive osteocytes (Figure 5E–H,M–P). This discrepancy in the presence of sclerostin-positive osteocytes between the newly formed and pre-existing bone seemed to be more evident in minimodeling-based bone formation than in remodeling-based bone formation; the three-dimensional reconstruction of the sclerostin-positive/sclerostin-negative osteocytes in the minimodeling site of the epiphyses in the ELD90 group demonstrated few sclerostin-reactive osteocytes in the minimodeling-induced new bone, while many sclerostin-reactive osteocytes in the underlying pre-existing bone were visually confirmed (Figure 5Q).

Therefore, we next analyzed the percentage of sclerostin-positive osteocytes in the minimodeling-induced new bone in the metaphyses and the epiphyses in all groups. In the metaphyses, around 20% of osteocytes were sclerostin-positive in the minimodeling-driven new bone; there was no significant differences in the percentage of sclerostin-positive osteocytes among the Sham, Vehicle, ELD30, and ELD90 groups (Figure 5R). In the epiphyses, however, the percentage of sclerostin-positive osteocytes in the minimodeling-driven new bone of the Sham group was more than 40%, and the ELD90 group exhibited significantly lower levels of sclerostin-positive osteocytes (Figure 5S). Thus, in the normal state, i.e., in the Sham groups, sclerostin appears to be synthesized in relatively larger amounts in the newly formed bone compared with the metaphyses. However, eldecalcitol treatment appears to reduce sclerostin synthesis in the minimodeling-induced new bone in the epiphyses rather than the metaphyses.

### 2.5. The Distribution of Osteocytic Lacunar Canalicular Network in Minimodeling-Induced New Bone

Since the geometrical distribution of the osteocytic lacunar canalicular network is important for maintaining bone quality [26,27], we have examined the distribution of osteocytes in the minimodeling-induced new bone, particularly in the epiphyses in the ELD90 group (Figure 6). Fluorescence microscopy revealed osteocytes parallel to the bone surfaces of focally convex bulges that featured calcein/tetracycline labeling (Figure 6A,B). Higher-resolution images obtained by confocal laser microscopy further showed that the osteocytes extended their cytoplasmic processes perpendicularly to the bone surface with calcein/tetracycline labeling (Figure 6C).

## 3. Discussion

We have previously reported that eldecalcitol increased bone mass by means of minimodeling-based bone formation and the suppression of osteoclastic bone resorption in not only an osteoporotic animal model but also osteoporosis patients [14,15,28]. Although it is not known in which region of the long bone minimodeling would be preferentially induced, this study has suggested that eldecalcitol could induce minimodeling-based bone formation in the epiphyses rather than the metaphyses of the tibiae and femora of OVX rats.

In the current study, the BV/TV and Tb.N of the metaphysis in the Vehicle group were significantly reduced, while the BV/TV and Tb.N of the epiphyses did not change in the Vehicle group compared with the Sham group (Figure 1). Therefore, compared with the metaphyses, the epiphyses may not be easily affected by stimulated osteoclastic bone resorption in ovariectomized rats, maintaining the bone volume and trabecular numbers to some extent. Bone turnover, that is, the combined activity of osteoclasts and osteoblasts, in the epiphyses appears to be lower than that in the metaphyses in a normal state. This is consistent with our findings of a lower numbers of osteoclasts, a smaller ALPase-positive area, and a higher percentage of sclerostin-positive osteocytes in the Sham epiphyses, compared with the metaphyses (Figure 5R,S).

We observed that the metaphyses exhibited predominantly minimodeling-based bone formation in rats receiving a high dose of eldecalcitol (90 ng/kgBW per day), though the value of minimodeling-based bone formation was not different from the ELD30 group. In contrast to the metaphyses, the epiphyses showed minimodeling-based bone formation at a lower dose of eldecalcitol (30 ng/kgBW per day) (Figure 4Q,R). This implies that the epiphyses of OVX rats would easily induce eldecalcitol-driven minimodeling. One may wonder why the epiphyses would show a relatively easier induction of minimodeling. We conjecture that the physiological functions of the epiphyses and metaphyses must be different: the epiphyseal bone volume may be maintained, being strong enough to resist the mechanical stress that may be transmitted from the knee joint, while the metaphyseal bone below the growth plate is the site of endochondral bone formation, enabling the longitudinal growth of the bone. As Frost and Jee’s colleagues [4,17] have proposed the idea that minimodeling would be induced by mechanical stress, we consider the possibility that the epiphyseal trabeculae, often exposed to mechanical stress, would promptly induce minimodeling to adapt to the strength and the direction of the mechanical stress.

Interestingly, minimodeling-induced bone shows a markedly reduced number of sclerostin-positive osteocytes, while there were many sclerostin-positive osteocytes in the pre-existing bone as shown in Figure 5. Sclerostin is known as an osteocyte-derived glycoprotein encoded by the Sost gene [29], suppressing osteoblastic activity [30,31,32,33]. Based on these reports, a neutral antibody to human sclerostin was developed, and the administration of this anti-sclerostin antibody induced a modeling-based bone formation independent of osteoclastic bone resorption in human patients [22,25] and in osteoporotic animal models [23,24]. As minimodeling-induced bone exhibits a markedly reduced number of sclerostin-positive osteocytes, eldecalcitol might suppress osteocytes to synthesize sclerostin. If so, eldecalcitol may activate the osteoblasts, though not as strongly as the anti-sclerostin antibody, to induce new bone formation through minimodeling.

In the metaphysis, the percentages of sclerostin-positive osteocytes in the minimodeling-induced bone did not change even after the eldecalcitol administration. In the epiphysis, in contrast, the percentage of sclerostin-positive osteocytes in the minimodeling-based bone was significantly diminished in the ELD90 group compared to the Sham group (Figure 5). Therefore, there may be some mechanism through which sclerostin synthesis would be inhibited in the epiphyses but not in the metaphyses upon eldecalcitol treatment. Since Robling et al. [34] have reported that mechanical stimulation reduces sclerostin expression in osteocytes, it may be possible that the epiphyses bearing the mechanical stress would reduce sclerostin synthesis in osteocytes, and therefore eldecalcitol could efficiently suppress sclerostin synthesis in such epiphyses in cooperation with the mechanical stress against the epiphyseal trabeculae.

We have previously reported that osteocytes embedded in mature bones, as opposed to those in immature bones such as metaphyseal primary trabeculae, express more abundant sclerostin [26,27] and that osteocytes show a regularly arranged distribution, extending their cytoplasmic processes perpendicularly to the bone surface [27]. Therefore, the osteocytes that were localized parallel to the bone surface and extending their cytoplasmic processes perpendicularly to the bone surface in this study may indicate that minimodeling-based bone was mature to some extent. If minimodeling is induced to bear mechanical stress, the resultant minimodeling-based bone must be strong enough to resist the mechanical stress. It seems necessary to explore the strength of the epiphyseal trabeculae reinforced by minimodeling-based bone formation as a result of eldecalcitol treatment in the future.

Finally, this study had some limitations stemming from its use of animal models: It remains unclear whether similar results would be replicated in older osteoporotic patients, as the current study utilized young rats with a higher rate of bone turnover compared to humans. Consequently, the external validity of this study, which used a rat model, is somewhat limited, suggesting that future research should involve osteoporosis patients. Additionally, it is imperative to assess the loads exerted on the epiphyseal and metaphyseal trabeculae of ELD-treated OVX rats, as previous research has suggested that mechanical loading may induce minimodeling [17]. Further investigations, including a stress analysis, may be necessary to elucidate the mechanisms of minimodeling-based bone formation, influenced by eldecalcitol, mechanical stress, and sclerostin synthesis.

In conclusion, our study indicates that ELD could preferentially induce minimodeling-based bone formation in the epiphyses rather than the metaphyses, and that ELD-driven minimodeling may be associated with the inhibition of sclerostin synthesis.

## 4. Material and Methods

### 4.1. Animals and Tissue Preparation

Sixteen-week-old female Wistar-Imamichi rats (purchased in the Institute for Animal Reproduction, Ibaraki, Japan) were randomized into four groups: sham-operated rats receiving vehicle (Sham group), ovariectomized (OVX) rats receiving vehicle (Vehicle group), OVX rats receiving ELD 30 ng/kgBW (ELD30 group) or 90 ng/kgBW (ELD90 group) (Table 1). ELD or vehicle (medium chain triglyceride, Nisshin Oillio, Tokyo, Japan) was orally administered five times per week for 12 weeks. An analgesic agent such as flunixin meglumine (2.5 mg/kg; DS Pharma Animal Health, Osaka, Japan) was subcutaneously administered on the day and 1 day after surgery. All rats subcutaneously received tetracycline (20 mg/kgBW) and calcein (6 mg/kgBW) 6 or 7 and 2 or 3 days before the fixation, respectively. This study was performed according to the experimental protocol approved by the Institutional Animal Care and Use Committee at Chugai Pharmaceutical Co., Ltd. (approved research proposal #15-109).

The rats, at 28 weeks of age and after vehicle or ELD administration, were anesthetized with isoflurane and blood was collected from the jugular vein, and then each organ was sampled. Left femora were used for detecting bone mineral density (BMD). Left tibiae were immersed in 70% ethanol and embedded in MMA resin to detect calcein/tetracycline labeling using a fluorescence microscope (Eclipse Ni, Nikon Instruments Inc., Tokyo, Japan) and a confocal laser microscope (LSM980 with Airyscan 2, ZEISS, Oberkochen, Germany). For histochemical analysis and bone histomorphometry, right tibiae were immersed in 4% paraformaldehyde diluted in 0.1 M phosphate buffer (pH 7.4) for 24 h at 4 °C. Then, they were decalcified with 10% EDTA and dehydrated in ascending concentrations of ethanol solutions prior to paraffin embedding. Sagittal sections were cut from paraffin-embedded right tibiae and MMA-embedded left tibiae. Sections were partitioned into two regions: the epiphyseal region above the growth plate and the metaphyseal region below the growth plate for histological assessment and statistical analyses of static bone histomorphometry (Table 1). Each region of these sections was photographed using light microscopy (Eclipse Ni, Nikon Instruments Inc.) and a digital camera (Nikon DS-Ri2 and NIS-Elements AR, Nikon Instruments Inc.).

### 4.2. Detection of BMD

BMD (mg/cm^2^) of the left femur was determined by dual-energy X-ray absorptiometry (DCS-600 EX, Aloka Co., Ltd., Tokyo, Japan). After obtaining the images, the long axis of the femur was divided into ten parts (equally divided into F1–F10, from the proximal to distal ends), and the BMDs of the whole femur (F1–F10) and the distal femur (F8–F10) were calculated.

### 4.3. Immunostaining for Tissue-Non-Specific Alkaline Phosphatase (ALPase), ED1, Cathepsin K, and Sclerostin

Dewaxed paraffin sections were examined for tissue-non-specific alkaline phosphatase (ALPase), cathepsin K, ED1, and sclerostin as reported elsewhere [35]. In brief, after the inhibition of endogenous peroxidases with methanol containing 0.3% hydrogen peroxidase for 30 min, dewaxed paraffin sections were pretreated with 1% bovine serum albumin (BSA; Serologicals Proteins Inc., Kankakee, IL, USA) in PBS (1% BSA-PBS) for 30 min. The sections were then incubated for 1–2 h at room temperature (RT) with rabbit polyclonal antisera against ALPase [36] diluted at 1:200 in 1% BSA-PBS, rabbit anti-cathepsin K (F-95, Daiichi Fine Chemical Co., Ltd., Toyama, Japan) at 1:300, mouse anti-rat ED1 (MCA341R, AbD Serotec, Oxford, UK) at 1:100, and goat anti-sclerostin (AF1589, R&D systems, Inc., Minneapolis, MN, USA) at 1:50. The sections reacted with antibodies to ALPase or cathepsin K were incubated with horseradish (HRP)-conjugated anti-rabbit IgG (P0399, DakoCytomation, Glostrup, Denmark). The sections reacted with antibodies to ED1 or sclerostin were incubated with HRP-conjugated anti-mouse IgG (61-6520, Chemicon International Inc., Temecula, CA, USA) for 1 h at RT, or HRP-conjugated anti-goat IgG (A201PS, American Qualex, San Clemente, CA, USA) for 1 h at RT, respectively. Diaminobenzidine tetrahydrochloride was used as a substrate to visualize HRP-conjugated immunoreactions. The sections reacted with sclerostin antibody were incubated with rabbit antisera to ALPase, and then with ALPase-conjugated anti-rabbit IgG (111-055-003, Jackson ImmunoResearch Laboratories, Inc., West Grove, PA, USA) at 1:100 at RT. The sections were incubated in a mixture of 2.5 mg of naphthol AS-BI phosphate (Sigma-Aldrich, St. Louis, MO, USA) and 18 mg of fast blue RR salt (Sigma-Aldrich) diluted in 30 mL of 0.1 M Tris-HCl buffer (pH 8.5) for 15 min at 37 °C. All the sections were counterstained with methyl green prior to observation under light microscopy.

### 4.4. Enzyme Histochemistry for Tartrate-Resistant acid Phosphatase (TRAPase)

Tartrate-resistant acid phosphatase (TRAPase) activity was detected as previously described [35]. In short, dewaxed paraffin sections were rinsed with PBS and incubated in a mixture of 2.5 mg of naphthol AS-BI phosphate (Sigma-Aldrich), 18 mg of red violet LB (Sigma-Aldrich) salt, and 100 mM L (+) tartaric acid (0.76 g) diluted in 30 mL of 0.1 M sodium acetate buffer (pH 5.0) for 15 min at 37 °C.

### 4.5. Bone Histomorphometry of BV/TV, Tb.N, Tb.Th, ALPase-Positive Osteoblastic Area, and the Analyses of TRAPase-Reactive, ED1-Positive, and Cathepsin K-Reactive Cells in Metaphyses and Epiphyses

We measured static parameters of bone histomorphometry such as bone volume per tissue volume (BV/TV, %), trabecular number (Tb.N, /mm), and trabecular thickness (Tb.Th, μm) in the metaphyses and epiphyses of right tibiae. We also performed histochemical evaluations for ALPase-immunoreactive osteoblastic area/tissue volume (%), the numbers of TRAPase-positive osteoclasts/bone surface (N/μm), ED1-positive cells/bone surface (N/μm), or cathepsin K-reactive osteoclasts/bone surface (N/μm) in the same regions of all groups. For the static parameters of bone histomorphometry and ALPase-positive area, as well as the numbers of TRAPase-positive cells, ED1-reactive cells, or cathepsin K-positive cells, the regions of interest (ROI) were defined as the metaphyseal region (1.3 mm × 2.0 mm area located 1.5 mm below the growth plate of the tibial metaphysis) and the epiphyseal region (1.0 mm × 1.4 mm area located at the center of the tibial epiphysis). Whenever possible, abbreviations were used and calculations were performed according to the recommendations of the ASBMR Histomorphometry Nomenclature Committee [37].

### 4.6. Quantification of the Frequency of Minimodeling-Based/Remodeling-Based Bone in Metaphyses and Epiphyses

Quantification of indices for the frequency of minimodeling-based/remodeling-based new bones was assessed in the ROI of epiphyses and metaphyses of all groups. The histological definition of minimodeling has been described elsewhere [4,6,7,14,17]. Briefly, new bones featuring a focally convex bulge and smooth continuous lines indicative of arrest lines on the underlying pre-existing bone were considered as minimodeling-based bone by employing MMA-embedded specimens [6,7,17]. On the contrary, flattened or slightly uplifted bones underlined with scalloped cement lines representing osteoclastic bone resorption were classified as remodeling-based bone [17]. If a new bone structure shows a focally convex shape but is underlined with scalloped cement lines, such bone was excluded from the category of minimodeling-based bone. The surface length of the areas showing minimodeling-based bones (encompassed by focally convex surfaces and smooth arrest lines) or remodeling-based bones (encompassed the flattened or slightly uplifted surface and scalloped cement lines) were measured, and divided by the whole length of the bone surface in the ROI of metaphyses and epiphyses (a 1.3 mm × 2.0 mm-region located 1.5 mm below the growth plate of the tibial metaphysis or a 1.0 mm × 1.4 mm-region located in the center of the tibial epiphyses).

### 4.7. Quantification of the Percentage of Sclerostin-Positive Osteocytes in Minimodeling-Based and Remodeling-Based Bone in Metaphyses and Epiphyses

The numbers of sclerostin-positive osteocytes and the total numbers of osteocytes located in the minimodeling-based and remodeling-based new bone were counted in the ROI of metaphysis or epiphysis for all groups (n = 6 per group). The average percentage of sclerostin-positive osteocytes in the minimodeling-based new bone in the ROI was statistically analyzed.

### 4.8. Statistical Analysis

Statistical analyses were performed by one-way ANOVA followed by Tukey–Kramer multiple comparisons test; however, the percentage of minimodeling-based versus remodeling-based bone in each group was statistically analyzed using Student’s *t*-test. The results are presented as the mean ± standard deviation (SD). Differences with *p* values lower than 0.05 were considered significant. The null hypothesis was established for each parameter in this study. The test statistic was calculated to determine whether the null hypothesis was rejected. The null hypothesis was rejected if the test statistic was less than the 5% significance level. Statistical analyses were performed on BellCurve for Excel (ver. 3.21, Social Survey Research Information Co., Ltd., Tokyo, Japan).

### 4.9. Three-Dimensional Reconstruction of Minimodeling-Based Bone Formation including Sclerostin-Positive/Negative Osteocytes

Twenty serial paraffin sections (approximately 100 μm) immunostained for sclerostin in the ELD90 group were used in the three-dimensional reconstruction of minimodeling-based bone formation. Minimodeling-based bone was identified as described above and distinguished from the pre-existing bone on two dimensional images of sclerostin immunohistochemistry. The minimodeling-based bone, the pre-existing bone, and sclerostin-positive/negative osteocytes were transferred to tracing papers and three-dimensionally reconstructed using the Fiji software (ver. 1.52h, National Institutes of Health, Bethesda, MD, USA) [38].

## Figures and Tables

**Figure 1 ijms-25-04257-f001:**
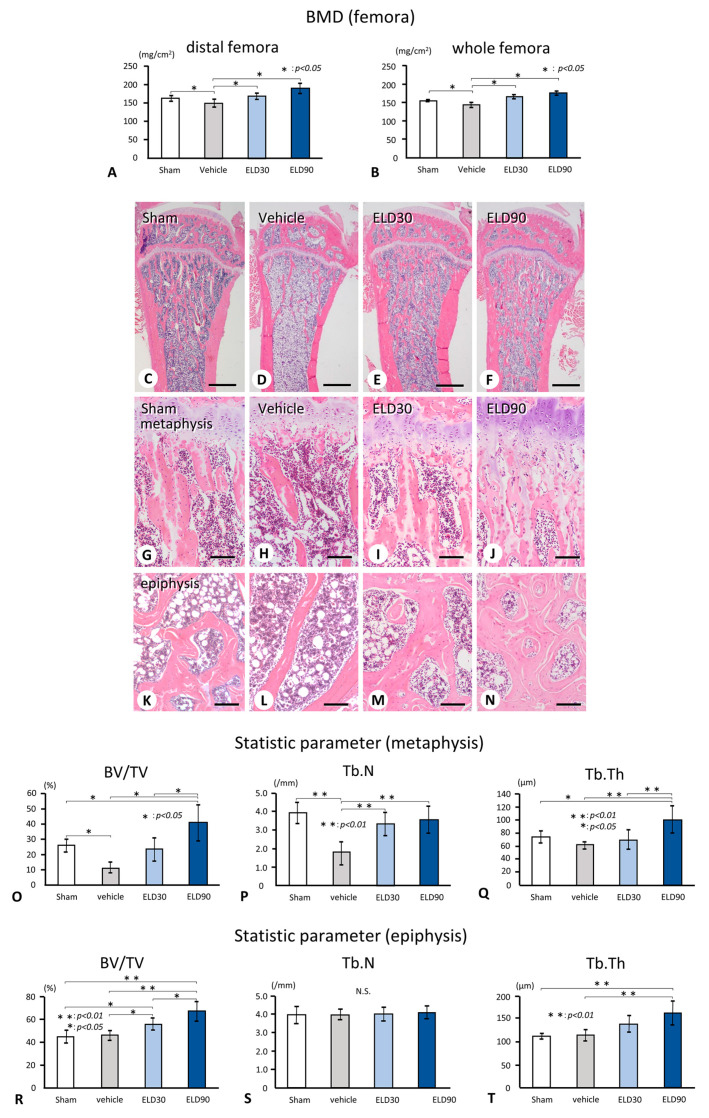
Histology, bone mineral density, and static characteristics of the femora and tibiae. Panels (**A**,**B**) show the values of distal/whole BMD in the femora of the Sham, Vehicle, ELD30, and ELD90 groups. The Vehicle groups have reduced values of distal and whole BMD in the femora compared to the Sham group (**A**,**B**). The values of femoral BMD in the ELD30 and ELD90 groups show significant differences compared to those of the Vehicle group. Panels (**G**–**J**) and (**K**–**N**) are highly magnified images of the metaphyses and the epiphyses of panels (**C**–**F**), respectively. H-E staining of the tibiae demonstrates the markedly reduced metaphyseal trabeculae in the Vehicle group (**D**,**H**,**L**) compared to the Sham group (**C**,**G**,**K**); however, the epiphyseal trabeculae still remain in the Vehicle group. ELD30 and ELD90 groups show the increased volume of metaphyseal trabeculae (**E**,**F**,**I**,**J**), and more stout trabeculae in the epiphyses (**M**,**N**). In the tibial metaphysis, BV/TV and Tb.N are markedly attenuated in the Vehicle group compared with the Sham group, but significantly increased in the ELD30 and ELD90 groups when compared with the Vehicle group (**O**,**P**). Although there is no significant difference in Tb.Th between the Sham and Vehicle groups, the ELD90 group demonstrates the highest value of Tb.Th in all groups (**Q**). In the epiphyses, there is no significant difference in BV/TV, Tb.N, and Tb.Th between the Sham and Vehicle groups (**R**–**T**). ELD administration increases BV/TV and Tb.N compared with the Vehicle group, although there is no significant difference in Tb.N among all the groups. All the values are presented as mean ± standard deviation. Values of *p* < 0.05 were considered significant. Bar, 1 mm (**C**–**F**), 100 μm (**G**–**N**).

**Figure 2 ijms-25-04257-f002:**
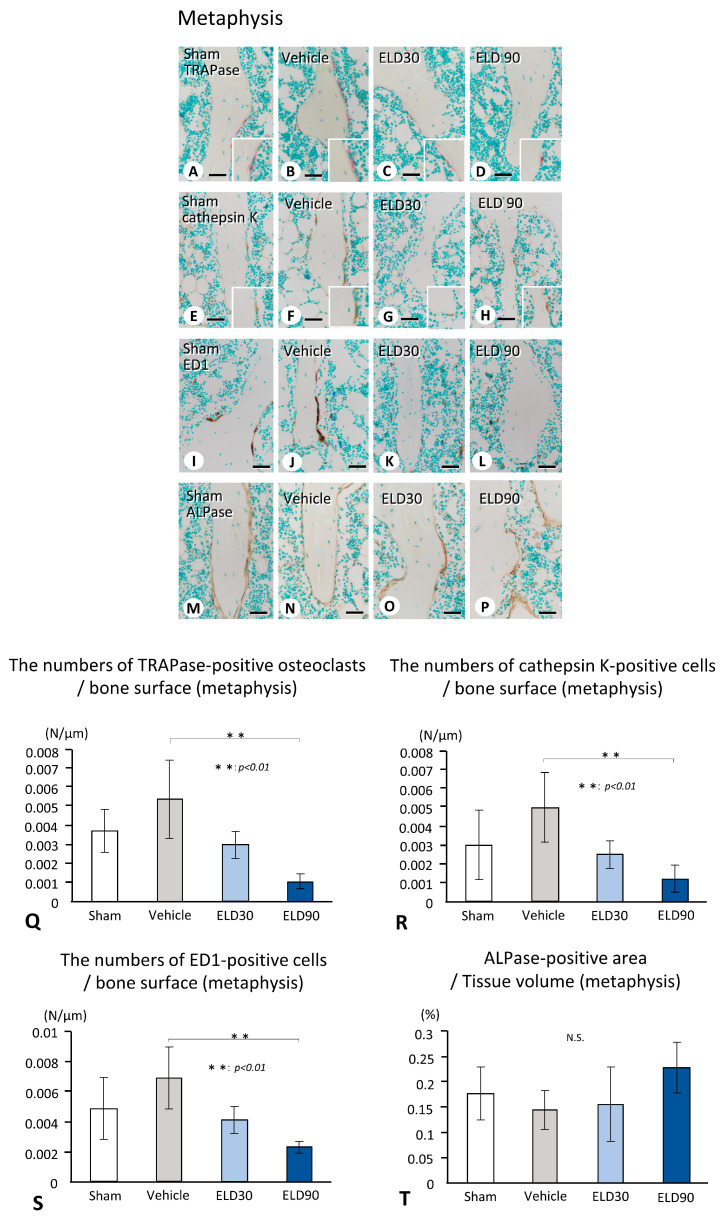
Histochemical analyses of TRAPase-reactive, cathepsin K-positive, ED1-immunoreactive, and ALPase-positive cells in the tibial metaphysis of Sham, Vehicle, ELD30, and ELD90 groups. Histochemical images of TRAPase-reactive (**A**–**D**), cathepsin K-positive (**E**–**H**), ED1-immunoreactive cells (**I**–**L**), and ALPase-reactive cells (**M**–**P**) are shown in the upper panels. When compared to Sham (**A**,**E**,**I**) and Vehicle (**B**,**F**,**J**) groups, ELD30 (**C**,**G**,**K**) and ELD90 (**D**,**H**,**L**) groups seem to have reduced numbers of TRAPase-reactive (**C**,**D**), cathepsin K-positive (**G**,**H**), ED1-immunoreactive (**K**,**L**) cells. On the contrary, some ALPase-reactive cells are observed in the Vehicle group (**N**); however, the ELD30 and ELD90 groups show intense ALPase-reactive cells (**O**,**P**). The statistical analysis results show a significant decrease in the numbers of TRAPase-reactive, cathepsin K-positive, or ED1-immunoreactive cells in the ELD90 group compared with the Vehicle group (**Q**–**S**). There is no significant difference in the statistical analyses of the ALP-positive area among all the groups (**T**). All the values are presented as mean ± standard deviation. Values of *p* < 0.05 were considered significant. Bar, 30 μm (**A**–**P**).

**Figure 3 ijms-25-04257-f003:**
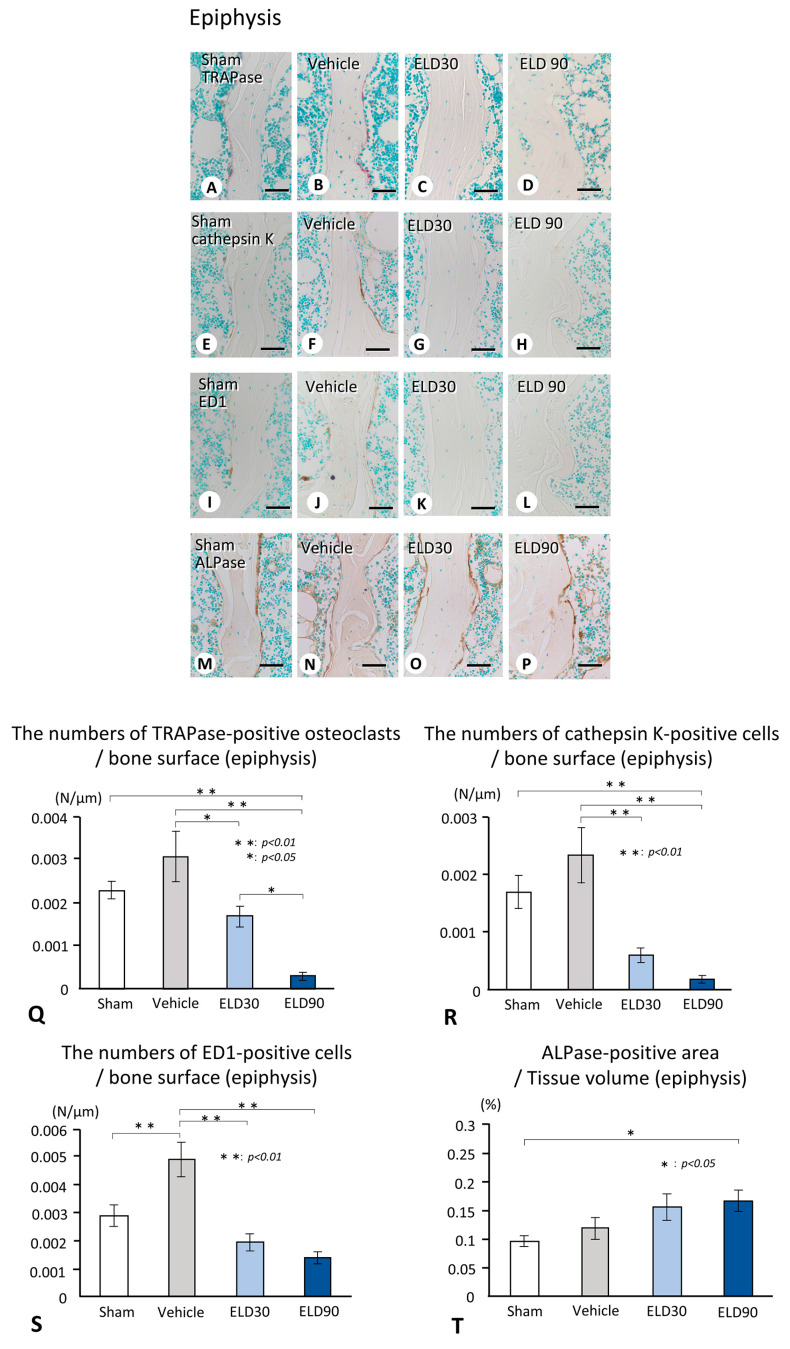
Histochemical analyses of TRAP-reactive, cathepsin K-positive, ED1-immunoreactive, and ALPase-positive cells in the tibial epiphyses of Sham, Vehicle, ELD30, and ELD90 groups. In the epiphysis, consistent with the metaphyses, ELD30 (**C**,**G**,**K**) and ELD90 (**D**,**H**,**L**) groups seem to have reduced numbers of the TRAPase-reactive (**C**,**D**), cathepsin K-positive (**G**,**H**), ED1-immunoreactive (**K**,**L**) cells when compared to Sham (**A**,**E**,**I**) and Vehicle (**B**,**F**,**J**) groups. Like the metaphyses, ALPase-reactive cells are weakly detected in the Vehicle group (**N**); however, the ELD30 and ELD90 groups show intense ALPase-reactive cells (**O**,**P**). Statistical analysis shows a significant increase in ED1-immunoreactive cells, but not in TRAPase-reactive and cathepsin K-positive cells (**Q**–**S**) in the Vehicle group. The numbers of cells positive for TRAPase, cathepsin K, and ED1 are significantly attenuated in the ELD30 and ELD90 groups when compared to the Vehicle group (**Q**–**S**). Unlike the metaphysis, the ALPase-positive area increases in the ELD 90 groups compared with the Vehicle group (**T**). All the values are presented as mean ± standard deviation. Values of *p* < 0.05 were considered significant. Bar, 30 μm (**A**–**P**).

**Figure 4 ijms-25-04257-f004:**
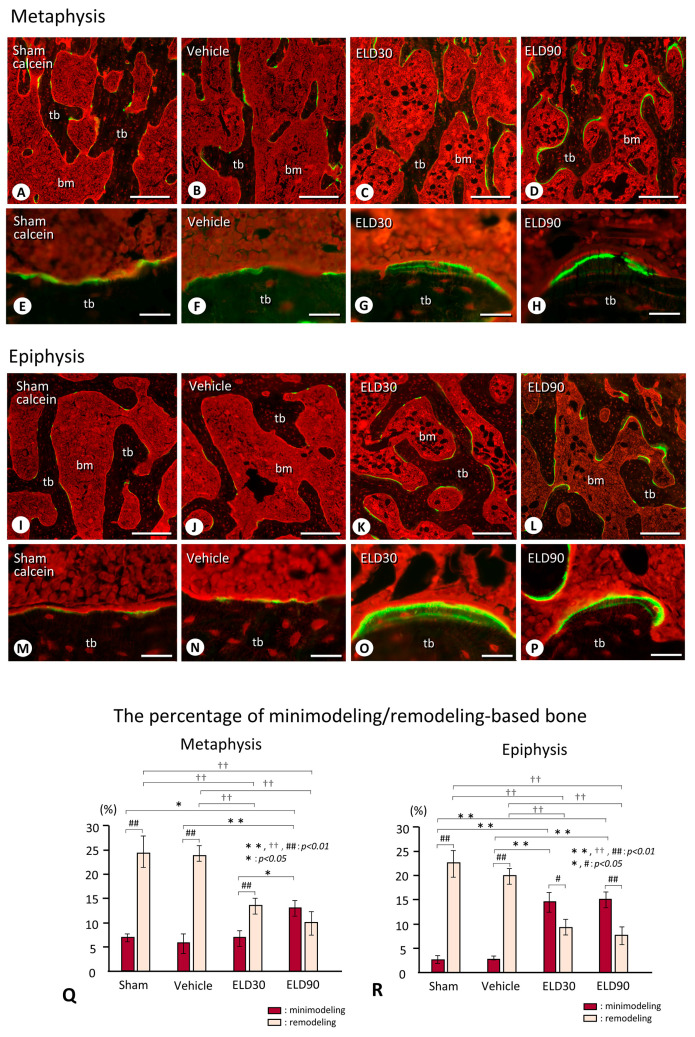
Calcein/tetracycline labeling and the percentage of the sites of minimodeling-based bone and remodeling-based bone in the metaphyseal and epiphyseal trabeculae of the Sham, Vehicle, ELD30, and ELD90 groups. Panels (**E**–**H**) and (**M**–**P**) are highly magnified images of the metaphyses (**A**–**D**) and the epiphyses (**I**–**L**) of the tibiae of the Sham (**A**,**E**,**I**,**M**), Vehicle (**B**,**F**,**J**,**N**), ELD30 (**C**,**G**,**K**,**O**) and ELD90 (**D**,**H**,**L**,**P**) groups, respectively. Note calcein/tetracycline labeling (green and yellow colored labeling) on the scalloped bone surface is indicative of remodeling-based bone formation in both the metaphyses (**E**,**F**) and epiphyses (**M**,**N**) of Sham (**E**,**M**) and Vehicle (**F**,**N**) groups. In ELD30 (**G**,**O**) and ELD90 (**H**,**P**) groups; however, calcein/tetracycline labeling can be seen mainly on focally convex bulges of bone surfaces, which is characteristic of minimodeling-based bone formation in both metaphyses (**G**,**H**) and epiphyses (**O**,**P**). The lower graphs show the percentage of minimodeling/remodeling-based bone in the metaphyses (**Q**) and epiphyses (**R**) of the Sham, Vehicle, ELD30, and ELD90 groups. The percentage of remodeling-based bone was higher than those of the modeling-based bone in both the metaphysis and epiphysis of the Sham and Vehicle groups. In the ELD30 group, the metaphyses show more remodeling-based bone than minimodeling-based bone, whereas the epiphyses reveal predominant minimodeling-based bone. Although the metaphyses showed a significant increase in minimodeling-based bone formation in the ELD90 group compared with ELD30 group, there was no difference between minimodeling-based bone formation and remodeling-based bone formation even in the ELD90 group. All the values are presented as mean ± standard deviation. **: *p* < 0.01, *: *p* < 0.05 (comparison of the percentage of minimodeling-based bone in all groups), ††: *p* < 0.01 (comparison of the percentage of remodeling-based bone in all groups), ##: *p* < 0.01, #: *p* < 0.05 (comparison between the percentage of minimodeling-based bone and remodeling-based bone in each group); tb: trabecular bone, bm: bone marrow. Bar, 200 μm (A–D,I–L), 30 μm (E–H,M–P).

**Figure 5 ijms-25-04257-f005:**
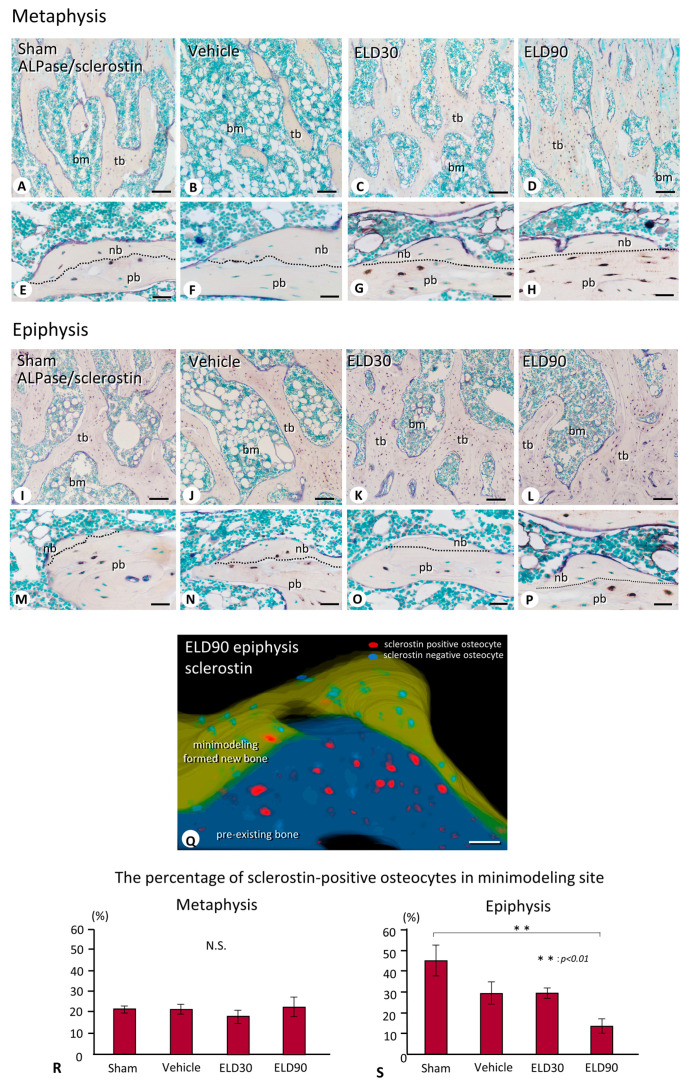
Double detection of ALPase and sclerostin and the percentage of sclerostin-positive osteocytes in minimodeling sites. Panels (**E**–**H**) and (**M**–**P**) are highly magnified images of the metaphyses (**A**–**D**) and the epiphyses (**I**–**L**) of the tibiae of the Sham (**A**,**E**,**I**,**M**), Vehicle (**B**,**F**,**J**,**N**), ELD30 (**C**,**G**,**K**,**O**), and ELD90 (**D**,**H**,**L**,**P**) groups, respectively. In the highly magnified images of metaphyses (**E**,**F**) and epiphyses (**M**,**N**) of Sham (**E**,**M**) and Vehicle (**F**,**N**) groups, remodeling-based bone, underlined with scalloped cement lines (the dotted line represents the boundary of newly formed bone and pre-existing old bone), hardly shows sclerostin-positive osteocytes (brown color), while pre-existing bone exhibits many sclerostin-positive osteocytes (**E**,**F**,**M**,**N**). Likewise, in the metaphyses (**G**,**H**) and epiphyses (**O**,**P**) of the ELD30 (**G**,**O**) and ELD90 (**H**,**P**) groups, minimodeling-based bone, underlined with smooth arrest lines (dotted lines indicate the boundary between the new bone and old bone), rarely shows sclerostin-positive osteocytes, while many sclerostin-positive osteocytes can be seen in the pre-existing bone. (**A,Q**) is a three-dimensional reconstruction representing the minimodeling-based bone featuring a large bulge of new bone (yellow-green color) on the smooth surface of pre-existing bone surface (dark blue color). The minimodeling-based bone contains many sclerostin-negative osteocytes (light blue color), whereas the pre-existing bone includes many osteocytes with sclerostin-immunoreactivity (red color). Panels (**R**,**S**) show the percentage of sclerostin-positive osteocytes in the minimodeling sites. In the metaphyses (**R**), the percentages of sclerostin-positive osteocytes are not significantly different between the Sham, Vehicle, ELD30 and ELD90 groups. In the epiphyses (**S**), the ELD90 group exhibits a significantly lower number of sclerostin-positive osteocytes compared to the Sham group. All the values are presented as mean ± standard deviation. Values of *p* < 0.05 were considered significant. pb: pre-existing bone; nb: newly formed bone. Bar, 200 μm (**A**–**D**,**I**–**L**), 30 μm (**E**–**H**,**M**–**P**), 50 μm (**Q**).

**Figure 6 ijms-25-04257-f006:**
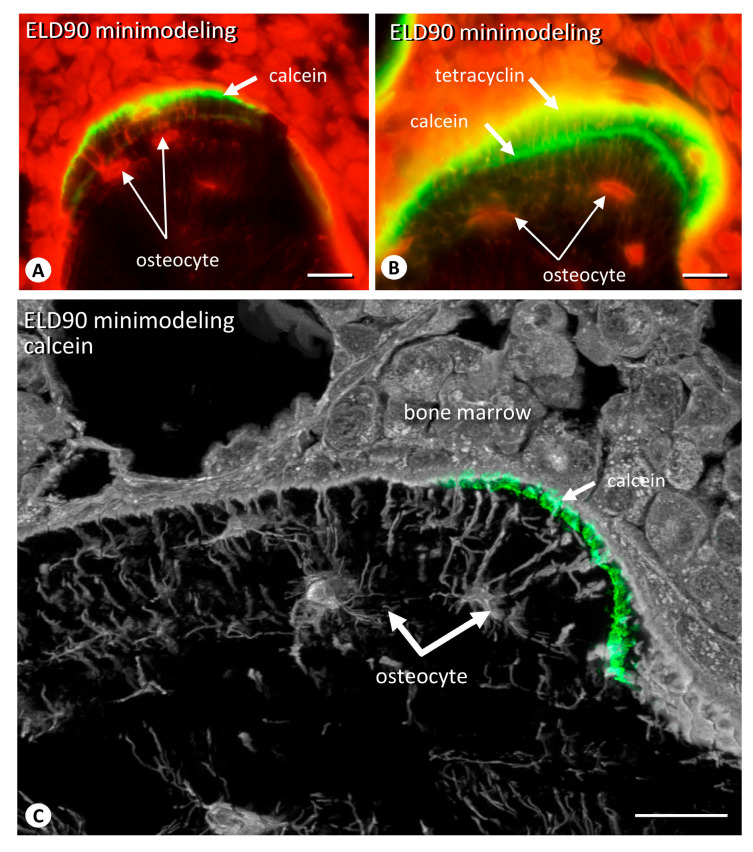
The network of the osteocytic lacunar canalicular system embedded in minimodeling-based bone. When observed at a higher magnification, fluorescent microscopy reveals calcein labeling (**A**) and calcein/tetracycline double labeling (**B**) indicative of newly formed minimodeling-based bone. Note, osteocytes are localized parallel to the bone surfaces. Confocal laser microscopic observation clearly shows osteocytes extending their cytoplasmic processes perpendicularly to the bone surface (**C**). Bar, 10 μm.

**Table 1 ijms-25-04257-t001:** Experimental group and the target region for analysis in this study.

Experimental Group	Operation	Treatment
Sham	Sham	Vehicle
Vehicle	OVX	Vehicle
ELD30	OVX	ELD 30 ng/kgBW
ELD90	OVX	ELD 90 ng/kgBW
**Tissue**	**Target region**	**Analysis item**
Left femur	Whole and distal	BMD
Right and left tibiae	Metaphysis	Histological assessment
	Epiphysis	Bone histomorphometry

## Data Availability

Data is contained within the article material.

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
