# Peer review of "Eldecalcitol Induces Minimodeling-Based Bone Formation and Inhibits Sclerostin Synthesis Preferentially in the Epiphyses Rather than the Metaphyses of the Long Bones in Rats"

_ijms, 2024, doi:10.3390/ijms25084257_

Round 1

Reviewer 1 Report

Comments and Suggestions for Authors

The authors showed that the eldecalcitol may preferentially induce bone formation based on mini-modeling in rat epiphyses and metaphyses areas. This paper is very well researched in great detail and the results are clear. Modifications to the following points would make it even easier for the reader to read.

・The 16 histopathology images in Fig.2 appear to be grouped together.

Wouldn't it be better to separate them into A-D, E-H, I-L, and M-P, or to lay them out in four rows of four images?

・The 16 histopathology images in Fig.3 appear to be grouped together.

Wouldn't it be better to separate them into A-D, E-H, I-L, and M-P, or to lay them out in four rows of four images?

・The 16 histopathology images in Fig.4 appear to be grouped together.

Wouldn't it be better to separate them into A-H, and I-P, or to lay them out in two rows of eight images?

・The 16 histopathology images in Fig.5 appear to be grouped together.

Wouldn't it be better to separate them into A-H, and I-P, or to lay them out in two rows of eight images?

・Is there any limitation in this study? Need to be noted within the discussion.

・The findings derived from the experimental results are not generalizable. The generalizability (external validity) of the study results should be discussed at the end.

Author Response

Our responses to reviewer #1

First, we would like to thank the reviewer for their invaluable suggestions regarding our study. We generally agree with your comments. Kindly find below our point-by-point responses to each of your comments. We believe that our manuscript has been significantly improved because of your valuable suggestions. Thank you very much for your important comments.

Reviewer’s general comments

The authors showed that the eldecalcitol may preferentially induce bone formation based on mini-modeling in rat epiphyses and metaphyses areas. This paper is very well researched in great detail and the results are clear. Modifications to the following points would make it even easier for the reader to read.

Reviewer’s specific comments #1

The 16 histopathology images in Fig.2 appear to be grouped together.

Wouldn't it be better to separate them into A-D, E-H, I-L, and M-P, or to lay them out in four rows of four images?

Our responses

Thank you for your suggestion. We agree with it and have separated the images of immunohistochemistry into four rows of four images each in the revised manuscript.

Reviewer’s specific comments #2

The 16 histopathology images in Fig.3 appear to be grouped together.

Wouldn't it be better to separate them into A-D, E-H, I-L, and M-P, or to lay them out in four rows of four images?

Our responses

Thank you for your suggestion. We have corrected the layout of Fig. 3 to that of revised Fig. 2.

Reviewer’s specific comments #3

The 16 histopathology images in Fig.4 appear to be grouped together.

Wouldn't it be better to separate them into A-H, and I-P, or to lay them out in two rows of eight images?

Our responses

Thank you for your comment. We have arranged the images of Fig. 4 in two rows, one for the epiphysis and the other for the metaphysis in revised Fig. 4.

Reviewer’s specific comments #4

The 16 histopathology images in Fig.5 appear to be grouped together.

Wouldn't it be better to separate them into A-H, and I-P, or to lay them out in two rows of eight images?

Our responses

Thank you for your comment. We have corrected the layout of Fig. 5 to that of revised Fig. 4.

Reviewer’s specific comments #5

Is there any limitation in this study? Need to be noted within the discussion.

Our responses

Thank you for your valuable comment. As suggested, we have described the limitations of this study in the Discussion section of the revised manuscript. We have also grouped our responses to this comment and the subsequent comments by the reviewer.

Discussion, Page 13, lines 591-600

Finally, this study had some limitations stemming from its use of animal models: It remains unclear whether similar results would be replicated in older osteoporotic patients, as the current study utilized young rats with a higher rate of bone turnover compared to humans. Consequently, the external validity of this study, which used a rat model, is somewhat limited, suggesting that future research should involve osteoporosis patients. Additionally, it is imperative to assess the loads exerted on the epiphyseal and metaphyseal trabeculae of ELD-treated OVX rats, as previous research has suggested that mechanical loading may induce minimodeling [17]. Further investigations, including stress analysis, may be necessary to elucidate the mechanisms of minimodeling-based bone formation, influenced by eldecalcitol, mechanical stress, and sclerostin synthesis.

Reviewer’s specific comments #6

The findings derived from the experimental results are not generalizable. The generalizability (external validity) of the study results should be discussed at the end.

Our responses

Thank you for your valuable comment. As highlighted by the reviewer, we acknowledge that the use of animal models limits the generalizability (external validity) of the findings obtained from the experimental results. Therefore, we included the statement "Consequently, the external validity of this study, which used a rat model, is somewhat limited, suggesting that future research should involve osteoporosis patients" in response to the reviewer's previous comment.

Discussion, Page 13, lines 591-600

Finally, this study had some limitations stemming from its use of animal models: It remains unclear whether similar results would be replicated in older osteoporotic patients, as the current study utilized young rats with a higher rate of bone turnover compared to humans. Consequently, the external validity of this study, which used a rat model, is somewhat limited, suggesting that future research should involve osteoporosis patients. Additionally, it is imperative to assess the loads exerted on the epiphyseal and metaphyseal trabeculae of ELD-treated OVX rats, as previous research has suggested that mechanical loading may induce minimodeling [17]. Further investigations, including stress analysis, may be necessary to elucidate the mechanisms of minimodeling-based bone formation, influenced by eldecalcitol, mechanical stress, and sclerostin synthesis.

We would like to thank you very much again for your excellent input as reviewer #1. Your comments have helped us to create a much-improved version of the original manuscript, which we hope is now suitable for publication.

Reviewer 2 Report

Comments and Suggestions for Authors

Dear Authors,

I carefully read your manuscript.

I think that it's well structured, well presented, the statistics is appropriate so I consider this work valuable of publication. Some points that I would address are:

- in the introduction, it would be better if you specify differences between epiphysis or metaphysis in preferential induction of minimodeling. Then, what differences between them could justify this preferential bone formation? 

- in M&M "the experimental protocol was approved by the Institutional ...." please insert the number or the code of the protocol

- why don't you consider a control group without ELD administration?

- No null hypothesis?

- please better clarify the groups formed in M&M as main groups and subgroups. I understand from the introduction that you want to differentiate between bone minimodeling on epiphysis or metaphyses while in the M&M section it seems that only two groups of comparison are present (ELD30 and ELD90).
Please assess this and then showed the results in the same order as M&M classification by group comparison. Then in the discussion, results have to be discussed in line with results presentation.

Author Response

Our responses to the reviewer #2

First, we would like to thank the reviewer for their invaluable suggestions on our study. We agree with your suggestions. Please kindly find below our point-by-point responses to each of your comments; the corresponding changes have been made in the main manuscript. We believe that our manuscript is now considerably improved owing to your insightful comments.

Reviewer’s general comments

I carefully read your manuscript.

I think that it's well structured, well presented, the statistics is appropriate so I consider this work valuable of publication. Some points that I would address are:

Reviewer’s specific comments #1

- in the introduction, it would be better if you specify differences between epiphysis or metaphysis in preferential induction of minimodeling. Then, what differences between them could justify this preferential bone formation?

Our responses

Thank you for your important comment. Previous studies have reported different frequencies of bone turnover and mechanical load between the epiphysis and metaphysis of the long bone in rats (Dempster et al, Bone. 1995; Hilliam et al, J. Bone. Miner. Res. 1995). Therefore, bone loss associated with bone remodeling after ovariectomy varies by each region of long bone (Baldock et al J. Bone. Miner. Res. 1998, Baldock et al J. Bone. Miner. Res. 1999). Similar to bone remodeling, we speculated that minimodeling induced on the trabeculae may also differ depending on the site of the long bones. We have described these points in the Introduction section of the revised manuscript and added the following references.

Introduction, Page 2, lines 70-79

It is likely that bone remodeling and minimodeling occur on the trabeculae. However, the frequency of bone turnover and mechanical loading in the trabeculae differ between the regions of the long bone, i.e., epiphysis or metaphysis [18, 19]. Because of the high bone turnover and low mechanical loading in the metaphyseal trabeculae compared to the epiphyseal trabeculae, the region of metaphysis undergoes marked bone loss after ovariectomy [20, 21]. As well as bone remodeling, we speculated that minimodeling induced on the trabeculae may also differ depending on the site of the long bones. However, which regions of ovariectomized long bones would preferentially induce minimodeling-based bone formation upon ELD treatment is unknown.

Reference

  1. Dempster, D.W.; Birchman, R.; Xu, R.; Lindsay, R.; Shen, V. Temporal changes in cancellous bone structure of rats immediately after ovariectomy. Bone. 1995, 16, 157–161.
  2. Hilliam, R.A.; Skerry, T.M. Inhibition of bone resorption and stimulation of formation by mechanical loading of the modeling rat ulna in vivo. J. Bone. Miner. Res. 1995, 10, 683–689. DOI: 10.1002/jbmr.5650100503
  3. Baldock, P.A.; Morris, H.A.; Need, A.G.; Moore, R.J.; Durbridge, T.C. Variation in the short-term changes in bone cell activity in three regions of the distal femur immediately following ovariectomy. J. Bone. Miner. Res. 1998, 13, 1451-1457. DOI: 10.1359/jbmr.1998.13.9.1451.
  4. Baldock, P.A.; Need, A.G.; Moore, R.J.; Durbridge, T.C.; Morris, H.A. Discordance between bone turnover and bone loss: effects of aging and ovariectomy in the rat. J. Bone. Miner. Res. 1999, 14,1442-1448. DOI: 10.1359/jbmr.1999.14.8.1442.

Reviewer’s specific comments #2

- in M&M "the experimental protocol was approved by the Institutional ...." please insert the number or the code of the protocol

Our responses

Thank you for your suggestion. We have added the approval number of this study to the M&M section of the revised manuscript.

Material and Methods, Page 13, lines 615-617

This study was performed according to the experimental protocol approved by the Institutional Animal Care and Use Committee at Chugai Pharmaceutical Co., Ltd. (approved research proposal #15-109).

Reviewer’s specific comments #3

- why don't you consider a control group without ELD administration?

Our responses

Thank you for your comment. The control group consisted of ovariectomized (OVX) rats receiving vehicle, termed “OVX group” in the first draft. However, it would be easier for the reader to understand it if it is described as the “Vehicle group” (administration with vehicle but not eldecalcitol). We have revised “OVX” to “Vehicle group” in the revised version of the manuscript.

Abstract, Page 1, lines 20-23

Sixteen-week-old female rats were divided into four groups: sham-operated rats receiving vehicle (Sham group), ovariectomized (OVX) rats receiving vehicle (Vehicle group), or ELDs (30 or 90 ng/kg BW, respectively; ELD30 and ELD90 groups).

Reviewer’s specific comments #4

- No null hypothesis?

Our responses

We thank you for your important comment. The null hypothesis was established for each parameter in this study. The test statistic was calculated to determine whether the null hypothesis was rejected. The null hypothesis was rejected if the test statistic was less than the 5% significance level. In this study, the null hypothesis was not rejected for several parameters. We have noted the null hypothesis test in the M&M section of the revised manuscript.

Material and Methods, Page 16, lines 720-723

4.8. Statistical analysis

Statistical analyses were performed by one-way ANOVA followed by Tukey-Kramer multiple comparisons test, however, the percentage of minimodeling-based versus remodeling-based bone in each group was statistically analyzed using student’s t-test. The results are presented as the mean ± standard deviation (SD). Differences with p values lower than 0.05 were considered significant. The null hypothesis was established for each parameter in this study. The test statistic was calculated to determine whether the null hypothesis was rejected. The null hypothesis was rejected if the test statistic was less than the 5% significance level.

Reviewer’s specific comments #5

- please better clarify the groups formed in M&M as main groups and subgroups. I understand from the introduction that you want to differentiate between bone minimodeling on epiphysis or metaphyses while in the M&M section it seems that only two groups of comparison are present (ELD30 and ELD90).

Please assess this and then showed the results in the same order as M&M classification by group comparison. Then in the discussion, results have to be discussed in line with results presentation.

Our responses

Thank you for your comment. In this study, we divided the female rats into four experimental groups; sham-operated rats receiving vehicle (Sham group), OVX rats receiving vehicle (Vehicle group), or OVX rats receiving ELD30 or 90 ng/kg BW (ELD30 and ELD90 groups), and the metaphyses and epiphyses of their long bones were analyzed. We have rewritten the M&M section and the Abstract in the revised manuscript and added Table 1 to make it easier for the readers to follow. The Results section presented the findings obtained from the metaphyses and epiphyses in each experimental group, and these results were discussed in accordance with the presentation of the results in the Discussion section. Minor revisions have been made to these sections to aid the reader's understanding.

Abstract, Page 1, lines 19-25

This study aimed to examine minimodeling-based bone formation between the epiphyses and metaphyses of the long bones of eldecalcitol (ELD)-administered ovariectomized rats. Six-teen-week-old female rats were divided into four groups: sham-operated rats receiving vehicle (Sham group), ovariectomized (OVX) rats receiving vehicle (Vehicle group), or ELDs (30 or 90 ng/kg BW, respectively; ELD30 and ELD90 groups). ELD administration increased bone volume and trabecular thickness, reducing the osteoclast number in both epiphyses and metaphyses of OVX rats. The Sham and Vehicle groups exhibited mainly remodeling-based bone formation in both regions.  

Material and Methods, Page 14, lines 628-631

4.1.        Animals and tissue preparation

…For histochemical analysis and bone histomorphometry, right tibiae were immersed in 4% paraformaldehyde diluted in 0.1 M phosphate buffer (pH 7.4) for 24 h at 4°C. Then, they were decalcified with 10% EDTA and dehydrated in ascending concentrations of ethanol solutions prior to paraffin embedding. Sagittal sections were cut from paraffin-embedded right tibiae and MMA-embedded left tibiae. Sections were partitioned into two regions: the epiphyseal region above the growth plate and the metaphyseal region below the growth plate for histological assessment and statistical analyses of static bone histomorphometry (Table 1). Each region of these sections was photographed by light microscopy (Nikon Ni, Nikon Instruments Inc., Tokyo, Japan) and a digital camera (Nikon DS-Ri2 and NIS-Elements AR, Nikon Instruments Inc.).

Material and Methods, Page 15, lines 683-687

4.5.        Bone histomorphometry of BV/TV, Tb.N, Tb.Th, ALPase-positive osteoblastic area, and the analyses of TRAPase-reactive, ED1-positive, and cathepsin K-reactive cells in metaphyses and epiphyses

For the static parameters of bone histomorphometry and ALPase-positive areas, and the numbers of TRAPase-positive cells, ED1-reactive cells, or cathepsin K-positive cells, and the regions of interest (ROI) were defined as the metaphyseal region (1.3 mm × 2.0 mm area located 1.5 mm below the growth plate of the tibial metaphysis) and the epiphyseal region (1.0 mm × 1.4 mm area located at the center of the tibial epiphysis).

 Again, we would like to express our sincere appreciation to Reviewer #2. We believe that your invaluable suggestions have helped us significantly improve our paper.
